# The Pitfall of White Blood Cell Cystine Measurement to Diagnose Juvenile Cystinosis

**DOI:** 10.3390/ijms24021253

**Published:** 2023-01-09

**Authors:** Tjessa Bondue, Anas Kouraich, Sante Princiero Berlingerio, Koenraad Veys, Sandrine Marie, Khaled O. Alsaad, Essam Al-Sabban, Elena Levtchenko, Lambertus van den Heuvel

**Affiliations:** 1Laboratory of Pediatric Nephrology, Department of Development and Regeneration, KU Leuven Campus Gasthuisberg, 3000 Leuven, Belgium; 2Department of Pediatrics, AZ Delta Campus Torhout, 8820 Torhout, Belgium; 3Laboratory of Inherited Metabolic Diseases/Biochemical Genetics, Cliniques Universitaires Saint-Luc, UC Louvain, 1200 Brussels, Belgium; 4Section of Histopathology, Department of Pathology and Laboratory Medicine, King Faisal Specialist Hospital and Research Centre, Riyadh 11533, Saudi Arabia; 5Section of Pediatric Nephrology, Department of Pediatrics, King Faisal Specialist Hospital and Research Centre, Riyadh 11533, Saudi Arabia; 6Department of Pediatrics, University Hospitals Leuven Campus Gasthuisberg, 3000 Leuven, Belgium; 7Department of Pediatrics, Division of Pediatric Nephrology, Amalia Children’s Hospital, Radboud University Medical Center, HB-6524 Nijmegen, The Netherlands

**Keywords:** cystinosis, cysteamine, cystine, nephropathic cystinosis

## Abstract

Cystinosis is an autosomal recessive lysosomal storage disease, caused by mutations in the *CTNS* gene, resulting in multi-organ cystine accumulation. Three forms of cystinosis are distinguished: infantile and juvenile nephropathic cystinosis affecting kidneys and other organs such as the eyes, endocrine system, muscles, and brain, and adult ocular cystinosis affecting only the eyes. Currently, elevated white blood cell (WBC) cystine content is the gold standard for the diagnosis of cystinosis. We present a patient with proteinuria at adolescent age and corneal cystine crystals, but only slightly elevated WBC cystine levels (1.31 ½ cystine/mg protein), precluding the diagnosis of nephropathic cystinosis. We demonstrate increased levels of cystine in skin fibroblasts and urine-derived kidney cells (proximal tubular epithelial cells and podocytes), that were higher than the values observed in the WBC and healthy control. *CTNS* gene analysis shows the presence of a homozygous missense mutation (c.590 A > G; p.Asn177Ser), previously described in the Arab population. Our observation underlines that low WBC cystine levels can be observed in patients with juvenile cystinosis, which may delay the diagnosis and timely administration of cysteamine. In such patients, the diagnosis can be confirmed by cystine measurement in slow-dividing cells and by molecular analysis of the *CTNS* gene.

## 1. Introduction

Nephropathic cystinosis (OMIM 219800) is a rare lysosomal storage disorder, caused by bi-allelic mutations in the *CTNS* gene (chr.17p13.2). *CTNS* encodes for the lysosomal cystine–proton transporter, cystinosin, and dysfunction results in the accumulation and crystallization of intra-lysosomal cystine in all organs [1,2].

Depending on the severity of the mutations in the *CTNS* gene, the age at presentation, and the severity of the clinical phenotype, three distinct clinical forms can be distinguished: the infantile nephropathic form, the juvenile nephropathic form, and the adult ocular non-nephropathic form [3]. Patients with infantile cystinosis are asymptomatic at birth but develop generalized proximal tubular dysfunction (also called renal Fanconi syndrome) by the age of 6–12 months, characterized by urinary loss of electrolytes, amino acids, glucose, low and intermediate molecular weight proteins, and water. Additionally, these patients present with failure to thrive, electrolyte imbalances, polyuria, polydipsia, hypocalcemia, hypokalemia, and the formation of vitamin D-resistant hypophosphatemic rickets [4,5]. Later on, the glomerular filtration rate deteriorates, eventually leading to the end-stage kidney disease (ESKD) [6,7]. The juvenile form of cystinosis affects less than 5% of the disease population, with the diagnosis usually made during adolescence and an overall milder phenotype [4,8]. These patients also experience a gradual decline of glomerular filtration rate (GFR), but ESKD usually develops at adult age [4,9]. The adult ocular non-nephropathic form of cystinosis is very rare, affecting less than 1% of the patients, and with a phenotype that is limited to the eyes [10]. Patients suffering from cystinosis are treated with the oral drug cysteamine, which depletes intra-lysosomal cystine levels and delays the progression to ESKD. However, a strict dosing schedule and accurate therapeutic follow-up is required [4]. Cysteamine should be administered immediately upon the diagnosis of nephropathic cystinosis, as any delay in treatment negatively impacts long-term renal outcome [11,12]. Patients with ocular cystinosis do not require systemic cysteamine, and their treatment is limited to focal cysteamine eye drops to treat corneal cystine crystals. Cysteamine eye drops are also indicated in patients with nephropathic cystinosis, as systemic cysteamine does not reach the cornea [13].

The current gold standard for the diagnosis of cystinosis is the detection of elevated cystine levels in white blood cells (WBC), measured by high-performance liquid chromatography or liquid chromatography–tandem mass spectrometry [14]. However, cystine measurement in WBCs, because of their short lifespan, may not always accurately reflect the total body cystine accumulation, which can hinder the accurate diagnosis and therapeutic monitoring. Furthermore, there is no standard cystine level for the diagnosis of cystinosis, with varying thresholds being used in different laboratories based on measurements in healthy patients [15,16]. Of note, among different types of WBC, cystine preferentially accumulates in polymorphonuclear (PMN) leukocytes (mainly in neutrophils) and not in lymphocytes. Hence, measuring cystine in mixed WBC preparations can lead to false low levels in young patients that have a predominance of lymphocytes in their differential blood count [17].

Here, we present a male patient in whom the diagnosis of nephropathic cystinosis was delayed because of WBC cystine levels below the diagnostic threshold utilized in our hospital. Measuring cystine in slow-dividing cells, such as skin fibroblasts and urine-derived kidney epithelial cells, demonstrated clearly elevated values as compared with the healthy cells. The diagnosis could also be confirmed by the genetic analysis that showed a homozygous mutation in *CTNS*, previously described in the Arab population [18].

## 2. Case

### 2.1. Clincal and Technical Examination

We studied a male patient of Arabian descent (weight: 83.1 kg, height: 167.8 cm, age: 21 years old) that was referred for a second opinion due to suspicion of juvenile cystinosis. At 12-years old, the patient complained of photophobia caused by corneal cystine crystals, and was subsequently diagnosed with ocular non-nephropathic cystinosis after genetic analysis of the cystinosin gene showed a homozygous mutation, but urine analysis revealed no anomalies at the time of diagnosis. Locally prepared cysteamine eye drops were prescribed at the age of 12 with improvement in photophobia. Commercially available cysteamine ophthalmic solution (Cystadrops, 0.37%) was started at the age of 17. The patient suffered from mild proteinuria from the age of 15 (total protein 0.59 g/24 h) with severe proteinuria from the age of 20 (1.09 g/24 h), but his WBC cystine levels, 0.4 and 0.77 nmol **½** cystine/mg protein (at the age of 14 and 15 years, respectively), precluded the diagnosis of nephropathic cystinosis.

Upon admission in our hospital at the age of 21, his kidney function was extensively studied (Table 1). At the time of analysis, the patient presented with proteinuria (0.7 g/g total protein/creatinine on spot urine) and had no systemic complaints. Blood tests showed normal serum electrolytes and bicarbonate levels, normal creatinine (0.91 mg/dL), and slightly elevated cystatin C (1 mg/mL). Creatinine-based eGFR (CKD-EPI) was 120 mL/min/1.73 m^2^ and eGFP cystatin C-based eGFR was 88 mL/min/1.73 m^2^ (Table 1). No aminoaciduria or glucosuria were observed.

A kidney biopsy was performed in 2021 and routinely processed and stained (Figure 1). Forty-one glomeruli were available for histopathological examination, of which two were globally sclerosed. Also observed was focal segmental mesangial expansion by mild-to-moderately increased mesangial matrix and mesangial cellularity, as well as a mild increase in the thickness of the glomerular capillaries. These were associated with focal segmental glomerular scarring in the form of obliteration of the glomerular capillaries, adhesion of the glomerular tufts to the Bowman’s capsule, and focal minimal glomerular hyalinosis in 3 glomeruli (Figure 1A). There was no evidence of glomerular proliferative activity, while hyperplasia, cytoplasmic foam changes (Figure 1B), and multinucleation of the glomerular visceral epithelial cells (podocytes) (Figure 1C,D) were readily identified. Sporadic renal proximal tubular epithelial cells exhibited pale, glassy eosinophilic cytoplasm and hydropic and cytoplasmic foam changes (Figure 1E), but no multinucleation of the tubular epithelial cells was identified. Only few interstitial foam cells were noted. Furthermore, the routine direct immunofluorescence study was negative for IgG, IgM, and IgA, as well as C3, C1q, and Kappa and Lambda light chains. No crystal deposition was found in the toludine blue-stained slides in the podocytes, tubular epithelial cells, or the interstitial space. Finally, meticulous structural examination showed focal effacement of the foot processes of the podocytes, but no cytoplasmic crystals were identified in the examined podocytes.

Neurological examination did not show signs of myopathy, with muscle strength being normal. Mild skeletal abnormalities, including pedes plani-valgi and genua valga, were observed. Further clinical evaluation showed normal puberal development (Tanner stage 5) and testis volume (R = 20 mL, L = 15–20 mL), with plasma testosterone, FSH and LH and THS and FT4 levels being within the normal range (Table 1). Finally, no spleno- or hepatomegaly was observed.

### 2.2. Cystine Measurement

Previous cystine measurement in white blood cells (WBC), performed in the country of origin, were normal to slightly elevated (0.90 nmol **½** cystine/mg protein), with some inter-laboratory differences, leading to the diagnosis of ocular non-nephropathic cystinosis. The value measured in our laboratory of PMN leukocytes was 1.31 nmol ½ cystine/mg protein, which resides in proximity to the heterozygote range (<1.00 nmol ½ cystine/mg protein), not allowing the diagnosis of nephropathic cystinosis (PMN cystine > 3.00 nmol ½ cystine/mg protein). Therefore, we measured cystine levels in other body cells, more specifically in skin fibroblasts, urine-derived proximal tubular cells, and urine-derived podocytes.

Patient urine was collected and used to generate urine-derived conditionally immortalized proximal tubular epithelial cells (ciPTECs) and podocytes (ciPODOs), as described previously [19]. Cells were differentiated at 37 °C for 10 days and 12 days, respectively. Cystine measurement in these cells revealed increased levels in ciPTECs—13.39 nmol (±0.69) ½ cystine/mg protein and ciPODOs—8.59 nmol (±0.62) ½ cystine/mg protein (Table 2, Appendix A). Additionally, fibroblasts were obtained from the skin biopsy as described previously [20], and the cystine levels in these cells (3.40 nmol ½ cystine/mg protein) also exceeded the threshold of heterozygotes (Table 2) [21,22].

### 2.3. Genetic Analysis

Genomic DNA was isolated from the blood of the patient and used to analyze the mutation in the *CTNS* gene. First, an allele-specific PCR was used to screen for the common 57 kb deletion including part of the *CTNS* gene. After no deletion was observed, the gDNA was further analyzed by PCR and DNA sequencing for the promotor region and the coding parts of the *CTNS*, as well as the intron–exon boundaries. Sanger sequencing revealed a homozygous missense mutation (c.530A > G; p.Asn177Ser) in the second transmembrane loop of the cystinosin protein (Figure 2). This mutation was also confirmed in the urine-derived proximal tubule epithelial cells and podocytes of the patient.

Inheritance was confirmed by subsequent genetic analysis on the mother and father of the patient, revealing the heterozygous c.530A > G (p.Asn177Ser) mutation in *CTNS* exon 8 in both parents.

With the clinical, genetic, and cystine-level analysis considered, we diagnosed the patient with juvenile cystinosis and initiated oral cysteamine therapy (delayed release cysteamine bitartrate (Procysbi, 2x 75 mg per day). Moreover, because of pronounced proteinuria, the patient was treated with an ACE-inhibitor (Lisinopril, 20 mg/day).

## 3. Discussion

Cystinosis is a rare disease, with an overall incidence rate between 1:100,000 and 1:200,000. However, incidence rates can be higher in certain sub-populations that are characterized by a high degree of consanguinity [4]. Until now, around 140 pathogenic mutations have been identified in the *CTNS* gene and there are several hotspot mutations present in specific populations. An Arabian cohort study, including 21 individuals from 13 families, illustrated an overall increase in prevalence, with all but one of the analysed individuals being homozygous for a respective mutation [18].

We evaluated a patient of Arabian descent, who presented with moderate proteinuria, but no signs of renal Fanconi syndrome. Initial WBC cystine measurements were below the threshold found in untreated patients, which precluded the diagnosis of nephropathic cystinosis and administration of systemic cysteamine. In our hospital, we measured cystine levels in slow dividing cells, such as fibroblasts and kidney epithelial cells, which were clearly elevated.

Upon genetic analysis, we identified a bi-allelic missense mutation in the *CTNS* gene (c.530A > G), residing in the second transmembrane region, leading to an amino acid substitution (pAsn177Ser). Notably, asparagine and serine are polar amnio acids, with asparagine playing an important role in the formation of the 3D structure, due to its capacity to make strong hydrogen bonds, and the formation of active enzyme sites. Moreover, asparagine is an important site of glycosylation, which is abundant in lysosomal membrane proteins such as CTNS [24]. Importantly, the mutation found in this patient (*CTNS* c.530A > G) has already been reported in an Arab patient who suffered from infantile cystinosis [18] and resides in a region where two other infantile cystinosis-associated mutations have been found (p.Asn177Thr and p.Pro200Leu) [25]. Additionally, the p.Asn177Thr variant was initially studied by Kalatzis et al., who transduced COS-7 fibroblasts with a pEGF-N1 plasmid, containing the *CTNS* p.Asn177Thr variant. The lysosomal localization (GYDQ) motif in the C-terminal tail was also deleted, leading to the expression of the cystinosin in the plasma membrane. Thereby, the uptake of a labelled cystine ([^35^S] L-cystine) by the cells could be measured at an acidic pH. For this, it was observed that the p.Asn177Thr mutation abolished cystine transport (0.70 ± 4.1% transport activity remaining). Moreover, this mutation is similar to the one presented by our patient, with threonine being another hydroxyl-containing amino acid, but having a methyl substituent on the β-carbon. Furthermore, this missense mutation has been found in two sisters with juvenile cystinosis as a compound heterozygote in combination with another unidentified mutation [25,26]. Therefore, the genetic analysis confirmed the diagnosis of juvenile cystinosis for our patient, and systemic cysteamine therapy was initiated.

This diagnosis, however, was initially obscured by a low WBC cystine level. Therefore, the patient was diagnosed with ocular non-nephropathic cystinosis and did not receive systemic cysteamine treatment, while early intervention might have delayed or prevented kidney damage. Most likely, the short lifespan of white blood cells (1–3 days [16,27]) does not allow sufficient cystine accumulation due to the residual function of the pAsn177Ser cystinosin. Therefore, we performed cystine measurement in long-living kidney cells. Urine-derived podocytes (non-dividing cells [28]), urine-derived proximal tubule cells (slow-dividing cells with increased mitosis upon injury [29,30]), and skin fibroblasts [31] all showed significantly elevated cystine levels, suggesting the diagnosis of juvenile nephropathic cystinosis.

In general, target WBC cystine levels in cystinosis patients treated with cysteamine are still a matter of debate. Mostly, levels within the range of heterozygotes carrying mutations in *CTNS* on one allele are considered to be safe, as heterozygotes do not develop symptoms of cystinosis, while reaching levels of healthy people requires high doses of cysteamine that are frequently not well-tolerated [32]. In our case, relatively low WBC cystine levels before the initiation of cysteamine make it uncertain which levels should be targeted for the therapeutic monitoring and dose adjustment. In this regard, an alternative biomarker of cystinosis, a macrophage enzyme chitotriosidase, reflecting whole-body cystine storage might be an interesting alternative [15]. However, 5% of the general population have a 24 bp duplication in the *CHIT1* gene that leads to a complete loss of function of the enzyme, which was also the case in our patient as no detectable chitotriosidase levels could be measured. Therefore, research towards new biomarkers to use for treatment monitoring is still of crucial importance. For this patient, we recommend therapeutic follow-up by means of assessment of the kidney phenotype itself.

Because of pronounced proteinuria, in addition to cysteamine, our patient was treated with an ACE-inhibitor (Lisinopril). While ACE-inhibitors are able to reduce proteinuria in cystinosis patients [33], it remains controversial whether this type of drug is beneficial for long-term kidney function survival in cystinosis [11,32]. More specifically, in a small single-center series, a better outcome of kidney function after long-term treatment with ACE-inhibitors has been demonstrated. However, a large multi-center longitudinal cohort study with over 450 patients did not show a decreased risk of CKD 5 after administration of ACE-inhibitors for at least 5 years [11]. However, because the kidney biopsy in our patient clearly showed signs of glomerular damage, we considered that the administration of ACE-inhibitors might reduce proteinuria and slow down kidney disease progression.

In conclusion, our observation underlines that low WBC cystine levels could preclude the diagnosis of juvenile nephropathic cystinosis in patients with missense mutations in the *CTNS* that allow for residual function of cystinosin. Therefore, we recommend that, in case of doubt, the diagnosis can be confirmed by cystine measurements in slow-dividing cells such as fibroblasts or kidney epithelial cells. Systemic cysteamine should be administered in these patients immediately after the diagnosis to prevent deterioration of kidney function and potential involvement of extra-renal organs. This case also shows that efforts must be made to ensure timely therapeutic intervention in the case of proteinuria development in patients originally diagnosed with ocular non-nephropathic cystinosis. If signs of kidney dysfunction appear in these patients, it must urge clinicians to perform further analysis, initiate appropriate treatment strategies, and make amendments to the original diagnosis.

## 4. Materials and Methods

### 4.1. Patient

A patient, presenting with proteinuria, was referred to our hospital after previous treatment in Saudi-Arabia. The patient was previously diagnosed and treated for ocular non-nephropathic cystinosis. Technical examinations were performed to assess the kidney function, and cystine levels were measured in the white blood cells according to the standard protocols from the University Hospitals Leuven. The patient gave an informed consent to publish this case report.

### 4.2. Generation of Conditionally Immortalized Cell Lines

PTEC and podocyte cell lines were generated from the urine of the patient and subsequently immortalized using the Simian Virus 40 large T antigen (SV40T) and human telomerase reverse transcriptase (hTERT), as described previously [19]. This process results in conditionally immortalized cells that grow at the permissive temperature of 33 °C, with grow-arrest and subsequent differentiation after translocation to the restrictive temperature of 37 °C. Cells were cultured in Dulbecco’s Modified Eagle’s Medium (DMEM) F-12 (Biowest, Nuaillé, France—cat.no. L0093-500) with 10% fetal bovine serum (FBS—Biowest, Nuaillé, France—cat.no S181B-500) and supplemented with 1.1% Penicillin and Streptomycin (Westburg, Leusden, The Netherlands—cat.no. DE17-602E) and 1% ITS (5 μg/mL insulin, 5 μg/mL transferrin and 5 μg/mL selenium) (Sigma, St. Louis, Missouri, United States—cat.no. I-1884). The PTEC medium was further supplemented with 10 ng/mL epidermal growth factor (Sigma, St. Louis, Missouri, United States—cat.no. E9644), 40 pg/mL triiodothyronine (Sigma, St. Louis, Missouri, United States—cat.no. T5516) and 36 ng/mL hydrocortisone (Sigma, St. Louis, Missouri, United States—cat.no. H0135).

### 4.3. gDNA Isolation from Blood

A whole blood sample was obtained from the patient, centrifuged, and both the red blood phase and interphase isolated. Red cell lysis buffer, comprising 2% Tris-HCl 1 M pH 7.5 (PlusOne Tris, GE Healthcare, Chicago, Illinois, United States-cat.no. 17-1321-01), 0.5% MgCl_2_ 1 M (Sigma, St. Louis, Missouri, United States-cat. no. M-2670) in water, was added and incubated for 15 min on ice, followed by a 15 min centrifugation step. This was done 3 times, resulting in a white blood cell pellet. The pellet was incubated overnight at 37 °C in SE buffer, comprising 2.5% 3 M NaCl (Sigma, St. Louis, Missouri, United States-cat.no. s7653) and 5% 0.5 M EDTA (pH 8.0, Sigma, St. Louis, Missouri, United States-cat.no. E-9884), supplied with 0.2 mg/mL of proteinase K (Invitrogen, Waltham, Massachusetts, United States-cat.no. AM2542) and 1% SDS (VWR, Leuven, Belgium-cat.no. 17-1313-01). After incubation, 5 M NaCl (Sigma, St. Louis, Missouri, United States-cat.no. S7653) was added and incubated for 15 min before centrifugation (15 min). DNA was precipitated from the supernatant by addition of 1 volume isopropanol (VWR, Leuven, Belgium-cat.no. 20842.312) and washed with 70% ethanol (VWR, Leuven, Belgium-1.00983.1000). DNA was pelleted by centrifugation (5 min), air dried, and resuspended in TE buffer (1% Tris-HCl (pH 7.5) and 0.2% 0.5 M EDTA (pH 8.0)).

### 4.4. gDNA Isolation from ciPTECs and ciPODOs

Patient-derived PTECs and podocytes were pelleted by means of trypsinization and resuspended in cell lysis buffer (10% Tris-HCl 1 M pH 8.5, 1% EDTA 0.5 M pH 8.0, 0.2% SDS, 4% NaCl 5 M in water) with 0.01 mg/mL proteinase K (Invitrogen, Waltham, Massachusetts, United States-cat.no. AM2542) by incubation at 55 °C for 2 h. The DNA was precipitated by addition of isopropanol (VWR, Leuven, Belgium-cat.no. 20842.312) to the supernatant, and the pellet was washed with 70% ethanol (VWR, Leuven, Belgium-cat.no. 1.00983.1000). The washed pellet was subsequently resuspended in TE buffer (1% Tris-HCl 1 M pH 7.5, 0.2% EDTA 0.5 M pH 8.0 in water).

### 4.5. Allele-Specific PCR

A mixture of 50 ng gDNA, 200 nM primers, and 50% GoTaqGreen (Promega, USA-cat.no. M7823) in water was prepared and subjected to 35 cycles of 30 sec at 94 °C, 54 °C, and 72 °C, respectively. Primers used were (1) a marker for the large deletion (forward—CTAACAGTATCACCGGAGTC, reverse—GGCCATGTAGCTCTCACCTC), and (2) a marker for the allele without the large deletion (forward—CTAGGGGAGCGTGTTAGCAT, reverse—TGTAAGACTGAGGCTGGAGC). As an internal control, Beta-actin was used (forward—GGCCAACCGCGAGAAGATGAC, reverse—CAGGGTACATGGTGGTGCCGC)

### 4.6. Sanger Sequencing of gDNA

Individual *CTNS* exons were amplified by means of a polymerase chain reaction with exon-specific primers (Table 3). Samples were prepared with 50% GoTaqGreen (Promega, Madison, Wisconsin, United States–cat.no. M7823) and 2% of each primer. Samples were incubated at 95 °C for 3 min, followed by 35 cycles of 30 sec at 95 °C, 60 °C and 72 °C, respectively. Next, DNA was purified with PureIT ExoZAP PCR CleanUp (Ampliqon, Odense, Denmark-cat.no. 75001) and purified products were prepared for sequencing through the Big Dye Terminator technology. Sequencing was done on the ABI 3100 sequence analyzer (Applied Biosystems, Waltham, MA, USA) and results were analyzed using SEQUENCE Pilot (JSI Medical Systems, Kippenheim, Germany).

### 4.7. Measurement of Intracellular Cystine in Cultured Kidney Cells

Cells were differentiated for 10 days (PTECs) and 12 days (podocytes) and collected in 5 mM of N-ethylmaleimide (NEM-Sigma, St. Louis, Missouri, United States- cat.no.04259) and 12% sulfosalicylic acid (SSA—Sigma, St. Louis, Missouri, United States-cat.no. S7422). Cells were lysed by centrifugation at high speed (13,000× *g* rpm, 10 min), and the supernatant was isolated and stored at −80 °C. Cystine measurement was performed at the Ospedale Pediatrico Bambino Gesù (PTECs and podocytes) and Cliniques universitaires Saint-Luc (UCLouvain-fibroblasts) by liquid chromatography and mass spectrometry analysis according to the hospital’s standard protocols.

In parallel, the cell pellet was dissolved in 0.1 M of Sodium Hydroxide (VWR, Leuven, Belgium–cat.no. 319511) and incubated overnight at 4 °C for full cell lysis. Protein concentration was determined via a Bicinchoninic Acid assay, using the commercial kit from Thermofisher Scientific (Waltham, Massachusetts, United States-cat.no. 23225), according to their protocol.

## Figures and Tables

**Figure 1 ijms-24-01253-f001:**
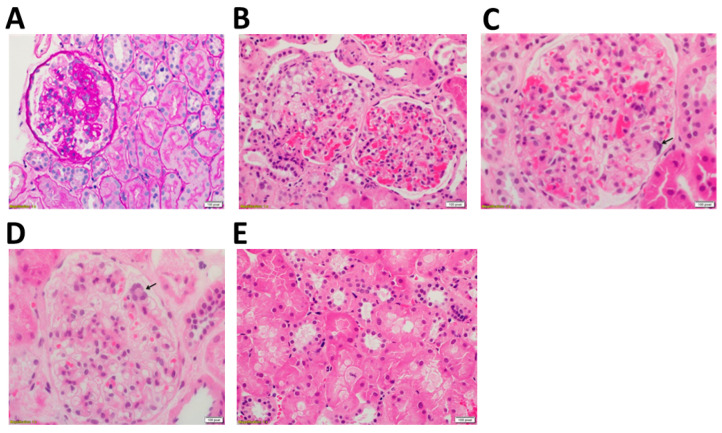
Histopathological manifestations in the patient. (**A**) Glomerulus shows signs of focal segmental glomerulosclerosis, which was likely due to podocytopathy, secondary to renal cystinosis. (**B**) Hyperplasia and cytoplasmic foam cell change in the glomerular visceral epithelial cells (arrow) were identified. (**C**,**D**) In the glomerular epithelium, few multinucleated podocytes were found (arrows). (**E**) Hydropic epithelial cells with pale eosinophilic cytoplasm and foamy change were observed in the kidney tubule. ((**A**) Periodic acid Schiff (PAS) 200×, (**B**) Hematoxylin and Eosin (H&E) 200×, (**C**–**E**) H&E 400×).

**Figure 2 ijms-24-01253-f002:**
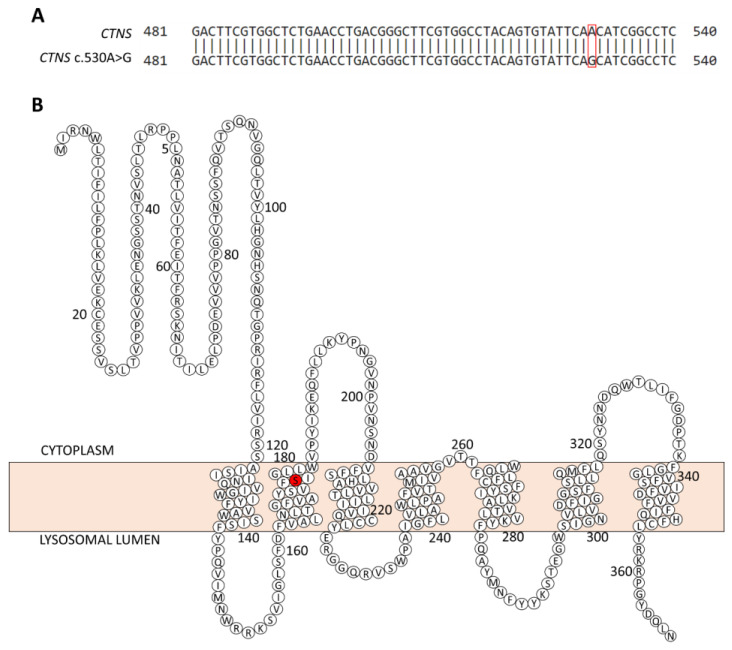
(**A**) A missense mutation (c. 530 A > G, red square) results in (**B**) an amino acid substitution (p. Asn177Ser, red) in the second transmembrane region of cystinosin. Adapted from [23].

**Table 1 ijms-24-01253-t001:** Clinical and technical examination of the patient at the last follow-up: blood and urine laboratory values shows pronounced proteinuria without renal Fanconi syndrome.

	Measurement at First Visit	Unit	Normal Range
Clinical examination			
Weight	83.1	kg	
Height	167.8	cm	
Blood Pressure	127/75	mmHg	
Blood			
Hemoglobin	14.0	g/L	14.0–18.0
Calcium	2.4	mmol/L	2.2–2.6
Phosphate	1.0	mmol/L	0.8–1.5
AP	137.0	U/L	40.0–130.0
PTH	35.0	ng/L	14.9–56.9
25-OH Vitamin D	23.9	µg/L	30.0–60.0
TSH	2.5	mIU/L	0.3–4.2
free T4	15.6	pmol/L	11.6–21.9
LH	4.7	IU/L	1.7–8.6
FSH	2.0	IU/L	1.2–7.7
Testosterone	696.0	ng/dL	300.0–1000.0
Urine			
Protein/creatinine	0.7	g/g	<0.2
Albumin/creatinine	602.0	mg/g	<30.0
Alpha-1 microglobulin/creatinine	17.9	mg/g	<11.7
TmP/GFR	1.0	mmol/L	0.8–1.2
CKD-EPI eGFR	120	ml/min/1.73 m^2^	>90.0
Cystatin C-eGFR	88	ml/min/1.73 m^2^	>90.0

**Table 2 ijms-24-01253-t002:** Cystine levels measured in different tissue cells.

	Case (nmol ½ Cystine/mg Protein)	Reference Values (nmol ½ Cystine/mg Protein) ^1^
PMN leukocytes	1.31	Normal: <0.20 [4] Heterozygotes: <1.00 [4] (Nephropathic) cystinosis: >3.00 [4]
Skin fibroblasts	3.40	Normal: <0.07 (mean) [22] Heterozygotes: <0.34 (mean) [22] (Nephropathic) cystinosis: >1.00 [21]
Podocytes (urine-derived)	8.59 ± 0.62	Normal: 5.01 ± 1.65 **
Proximal tubular epithelial cells (urine-derived)	13.39 ± 0.69	Normal: 2.04 ± 0.68 **

^1^ Reference values were based on literature references and own control ciPTEC and ciPODO data (**).

**Table 3 ijms-24-01253-t003:** Primers used for PCR amplification of the *CTNS* coding exons and the intron–exon boundaries.

Exon ^1^	Sequence
*CTNS* EX 3 FOR	AGC TGA TTC AAC ATT CCC CTG
*CTNS* EX 3 REV	TAG CCA CCA TTT CCC TCT TTA C
*CTNS* EX 4 FOR	TGT CAT TGA TTT GGG TCC TTC C
*CTNS* EX 4 REV	TAG GGC TTG TCT TAC AGG TA
*CTNS* EX 5 FOR	GAT CTC ACT GTC CAG CTT CT
*CTNS* EX 5 REV	TCC CTA CCC ATC CGT TAA G
*CTNS* EX 6 FOR	GCG GGG TCC TCG GTA ACT G
*CTNS* EX 6 REV	GGC CCC CTT CTT GTC ACG
*CTNS* EX 7 FOR	CTT CAG AAG CCC AGC CTC AGC
*CTNS* EX 7 REV	CGA GAG AGC CTG CAC ATA CG
*CTNS* EX 8 FOR	CCC TGC CCT GTC TTG TCC
*CTNS* EX 8 REV	CAG AGA TGT AGG GCA GGC AA
*CTNS* EX 9 FOR	CCT CAC CAC CCA GCT TCT CC
*CTNS* EX 9 REV	GTG GCG GGT GTT GGC TG
*CTNS* EX 10 FOR	GGC CTC TGT GTG GGT CC
*CTNS* EX 10 REV	GGC CAT GTA GCT CTC ACC TC
*CTNS* EX 11 FOR	GCC CTC CGT CTG TCT GTC CG
*CTNS* EX 11 REV	GCC CGA TGC CCC AGC CGC
*CTNS* EX 12 FOR	GCC AAC CTA ACA CCA GCT TC
*CTNS* EX 12 REV	AGA GGC TGG GTA CAC TGG GT

^1^ FOR = forward primer, REV = reverse primer.

## Data Availability

The clinical data presented in this study are available on request from the corresponding author. The data are not publicly available due to patient privacy. Cystine measurements are available in Appendix A.

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
