# Peer review of "The Pitfall of White Blood Cell Cystine Measurement to Diagnose Juvenile Cystinosis"

_ijms, 2023, doi:10.3390/ijms24021253_

Round 1
Reviewer 1 Report
This is an interesting case report about a patient with nephropatic cystinosis and normal WBC cystine levels, and how the measurement of cystine in slow-dividing cells is high, in agreement with the molecular diagnosis of juvenile cystinosis. The scientific method is sound and clearly explained, and it is of high interest as it is a disease whose progression can be partly delayed with a prompt treatment, so correct diagnosis is very important.
Author Response
We thank the reviewer for his/her positive comments and interest in our paper.
Reviewer 2 Report
“The pitfall of white blood cell cystine measurement to diagnose juvenile cystinosis” by Bondue et al., is an interesting case report that highlights the controversial levels of the WBC cystine levels which is the current gold standard for the diagnosis of the ultraorphan rare disease, cystinosis. Here are few comments:
- - Was the patient diagnosed in his country of origin as ocular cystinosis based on identification of ocular cystine crystals and the absence of any urinary sediment? Kindly clarify if the diagnosis was genetically confirmed then at initial diagnosis?
- - Page 2. Line 92-94 “The patient suffered from mild proteinuria from the age of 15 (total protein 0.59 92 g/24hrs) with severe proteinuria from the age of 20 (1.09g/24hrs), but his WBC cystine 93 levels, 0.4 and 0.77 nmol/ ½ cystine/mg protein (at the age of 14 and 15 years, respectively), 94 precluded the diagnosis of nephropathic cystinosis.” Kindly elaborate if there had been any efforts to diagnose the cause of the proteinuria and/or any therapy during those 6 years. In other words, was the progression of the renal disease in this patient modified by any antiproteinuric therapy?
- - Page 6, Line 175 Authors are advised not to use trade names. Change the trade name of the cysteine depleting therapy “Procysbi” to its chemical name “delayed release cysteamine bitartrate”.
- - The authors need to emphasize the gravity of the development of proteinuria in presumably ocular cystinosis patients even with WBC cysteine levels below the diagnostic levels. If anything, this case report highlights that proteinuria in presumably ocular cystinosis patient is alarming and should be taken seriously. Juvenile cystinosis should be ruled out in any ocular cystinosis patient with proteinuria by other measures, as cystine measurements in slow dividing cells such as fibroblasts or kidney epithelial cells, before labeling the patient as ocular cystinosis to ensure timely initiation of systemic cysteine depleting therapy to prevent/retard decline of kidney functions.
